# The Contribution of Vascular Proteoglycans to Atherothrombosis: Clinical Implications

**DOI:** 10.3390/ijms241411854

**Published:** 2023-07-24

**Authors:** Amelia Drysdale, Amanda J. Unsworth, Stephen J. White, Sarah Jones

**Affiliations:** 1Department of Life Sciences, Manchester Metropolitan University, Manchester M1 5GD, UK; a.drysdale@mmu.ac.uk (A.D.); a.unsworth@mmu.ac.uk (A.J.U.); 2Faculty of Medical Sciences, The Medical School, Newcastle University, Framlington Place, Newcastle upon Tyne NE2 4HH, UK; steve.white3@ncl.ac.uk

**Keywords:** proteoglycan, platelet, atherosclerosis, plaque rupture, plaque erosion, myocardial infarction, thrombosis, extracellular matrix

## Abstract

The vascular extracellular matrix (ECM) produced by endothelial and smooth muscle cells is composed of collagens and glycoproteins and plays an integral role in regulating the structure and function of the vascular wall. Alteration in the expression of these proteins is associated with endothelial dysfunction and has been implicated in the development and progression of atherosclerosis. The ECM composition of atherosclerotic plaques varies depending on plaque phenotype and vulnerability, with distinct differences observed between ruptured and erodes plaques. Moreover, the thrombi on the exposed ECM are diverse in structure and composition, suggesting that the best antithrombotic approach may differ depending on plaque phenotype. This review provides a comprehensive overview of the role of proteoglycans in atherogenesis and thrombosis. It discusses the differential expression of the proteoglycans in different plaque phenotypes and the potential impact on platelet function and thrombosis. Finally, the review highlights the importance of this concept in developing a targeted approach to antithrombotic treatments to improve clinical outcomes in cardiovascular disease.

## 1. Introduction

Myocardial infarction (MI) remains a leading cause of death globally [1]. In most cases, MI is caused by thrombotic occlusion of coronary arteries triggered by the rupture or erosion of an atherosclerotic plaque. Typically, plaque rupture occurs on macrophage and lipid-rich plaques with thin fibrous caps. In contrast, erosion affects plaques with thickened fibrous caps abundant in smooth muscle cells (SMCs) and extracellular matrix (ECM) proteins, with a deep-seated core and few resident leukocytes [2,3,4,5,6]. Both fibrous cap rupture and endothelial erosion can promote the formation of occlusive thrombi; however, the composition and morphology of the thrombi are distinct, suggesting that different pathological mechanisms are responsible. Despite the clear differences, current therapeutic strategies are one-size-fits-all, with no patient stratification.

Tailoring antiplatelet therapy to address the different mechanisms driving thrombus formation between ruptured and eroded plaques could provide a potential avenue to improve antithrombotic efficacy and reduce iatric bleeding events. Recent advances in intravascular imaging with optical coherence tomography (OCT) have now made it possible to distinguish rupture from erosion-prone plaques [7,8], opening the possibility of precision medicine and treating patients based on plaque phenotype. However, a better molecular understanding of the distinct mechanisms underpinning thrombus formation at sites of plaque rupture and erosion is required to facilitate this.

Seminal work by Virmani and colleagues characterizing the exposed substrates at sites of plaque rupture or erosion has identified distinct differences in the proteoglycan composition and abundance between the two etiologies. Decorin and biglycan are prominent at the luminal interface in ruptured plaques, while versican and hyaluronan are characteristic of eroded plaques [9,10,11]. Individual proteoglycans have unique effects on cell behavior and platelet responses, and the contribution of each to atherothrombosis will be the focus of this review.

## 2. Atherosclerosis

Atherosclerosis is characterized by the sub-endothelial accumulation of fat and fibrous material in the vascular wall, resulting in intimal thickening (Figure 1). Initiation and progression of atherosclerosis are primarily influenced by the expansion of the intima at sites of disturbed flow [12], followed by chronic accumulation and oxidation of low-density lipoprotein cholesterol (LDL-C) [13,14]. This is accelerated by activation and dysfunction of the endothelium caused by risk factors associated with atherosclerosis (e.g., hypertension, smoking, hyperlipidemia). The bioavailability of nitric oxide is reduced, with a concomitant upregulation of adhesion molecules (e.g., P-selectin, ICAM-1, VCAM-1) and increased vascular permeability [15]. Invading monocyte-derived macrophages and SMC internalize oxidized LDL and become foam cells, forming an intimal xanthoma (fatty streak) [10,16]. The fibrous cap in early lesions consists of SMCs and a variety of ECM components overlying a necrotic core of extracellular lipids and cell debris encircled by macrophages and T-cells [17,18]. Later lesions are classified based on plaque morphology and stability, the latter of which indicates whether a plaque is prone to rupture or erosion (Figure 2).

### 2.1. Plaque Rupture

Plaque rupture involves a biomechanical failure of the fibrous cap, exposing blood to the necrotic core. An imbalance in collagen production and degradation progressively weakens the fibrous cap until it is unable to withstand the mechanical forces of arterial pulsatile blood flow. Matrix metalloproteinases (MMPs) released predominantly from leukocytes degrade the ECM, which cannot be replaced due to SMC rarefaction through senescence and apoptosis. Plaque rupture exposes tissue factor (TF), von Willebrand factor (vWF), collagens, proteoglycans, and lipids, initiating platelet adhesion, activation, and coagulation [10,19]. Platelet glycoprotein (GP)VI and GPIb bind to exposed collagen and vWF, activating platelets and inducing a conformational change of integrins α_IIb_β_3_, α_2_β_1_, and α_V_β_3_. This allows ligand binding to fibrinogen, vWF, and other ECM proteins, promoting aggregation and thrombus formation [20]. The release of soluble mediators, including ADP and thromboxane A2 (TXA2), amplifies the platelet response resulting in rapid platelet recruitment and thrombus growth. Thrombus stabilization occurs through activation of the coagulation cascade, stimulated by TF that is exposed following plaque rupture and facilitated by platelets exposing phosphatidylserine (PS) [21]. Thrombin generated by the coagulation cascade contributes to platelet activation via protease-activated receptor (PAR)-1 and PAR-4, and cleaves fibrinogen, creating a fibrin mesh to consolidate thrombus formation. The growing fibrin-rich thrombus traps circulating erythrocytes, which aggregate and promote vessel occlusion. As a result, the final thrombus characteristic of plaque rupture has a platelet-rich head and a fibrin- and erythrocyte-rich tail [10,16].

### 2.2. Plaque Erosion

The vascular environment in plaque erosion, in contrast, has few inflammatory cells, a smaller, deeper seated necrotic core, and a higher density of myeloperoxidase-positive cells and neutrophil extracellular traps (NETs) [10,11,22,23]. Endothelial detachment observed in plaque erosion has been linked with amplified Nrf2-regulated gene expression [24] and toll-like receptor (TLR) promoted apoptosis, augmented by NETosis, resulting in exposure of the underlying ECM and SMCs to circulating platelets [11,17]. Histones released from NETs activate platelets and exposed vascular proteoglycans interact with platelets to promote local thrombosis. The resulting thrombus is platelet-rich and less frequently occlusive than thrombi resulting from rupture [11], suggesting differences in the mechanisms of thrombus formation. In addition, histopathology studies of eroded plaques frequently show evidence of thrombus remodeling, indicating that fatal erosions may have occurred 3–7 days before clinical presentation/death, allowing time for thrombus remodeling—again demonstrating a divergence in thrombus formation from plaque rupture [25,26].

### 2.3. Current Treatment for Atherothrombosis

Current preventative treatments for atherothrombosis largely target pathways of soluble platelet agonists, ADP, TXA_2_, and thrombin (Figure 3), which amplify thrombus growth following release from activated platelet. Dual antiplatelet therapy (DAPT), a combination of aspirin and thienopyridines such as clopidogrel, is the most common approach, reducing the risk of major vascular events by over 10% [27]. Aspirin inactivates platelet cyclo-oxygenase (COX)-1, preventing thromboxane A2 synthesis [28], and thienopyridines block ADP P2Y12 receptor activity. Treatment efficacy is, however, limited by high inter-individual variability, and the risk of recurrent events/death after 12 months remains at around 20% [16]. Bleeding also represents a significant problem, particularly in older patients and those with co-morbidities.

Current research aims to optimize a combination of novel antiplatelet drugs, including potent P2Y12 receptor blockers such as prasugrel and ticagrelor, and PAR-1 receptor antagonists, such as vorapaxar. Advances have also been made in the development of novel antiplatelet drugs which disrupt platelet collagen interactions, with two agents entering phase II clinical trials. Revacept is a dimeric fusion protein comprising the extracellular domain of GPVI and the human Fc-fragment [29]. It competes with endogenous GPVI receptors to bind collagen exposed in atherosclerotic lesions. Glenzocimab is a humanized fab fragment that binds GPVI blocking its interaction with collagen [30]. Understanding the distinct contributions of matrix proteoglycans to platelet activation and thrombus formation in both ruptured and eroded plaques may identify further novel targets aimed at disrupting platelet and ECM interactions and could facilitate a more stratified approach to antiplatelet therapy with improved efficacy and reduced incidence of bleeding.

## 3. Proteoglycans and Atherothrombosis

Vascular ECM is composed of collagens and glycoproteins, including proteoglycans and glycosaminoglycans (GAGs), which regulate vascular structure and function. The core GAGs found in the vascular wall are chondroitin sulphate (CS), dermatan sulphate (DS), heparan sulphate (HS), and keratan sulphate (KS), which may attach to a core protein to form a proteoglycan [31]. Vascular proteoglycans and GAGs have been shown to contribute to a variety of cardiovascular disease processes, including heart failure and vascular stiffness, through sodium buffering and modulation of endothelial function [32,33,34]. Proteoglycans also exhibit important roles in the development and progression of atherosclerosis by regulating cell behavior (Table 1) [35,36,37,38,39], promoting lipid retention, and modulating elastic fiber assembly and inflammation [40,41,42]. Interspecies variation in ECM composition renders current animal models largely unrepresentative of human atherothrombosis [31], particularly in cases of plaque erosion, where vascular proteoglycans are abundant at the plaque–thrombus interface and contribute to thrombus formation [10]. These species differences may go some way in explaining the lack of efficacy or success of current therapies targeting ACS and the limitations in the development of new therapies that specifically target plaque erosion.

### 3.1. Perlecan

Perlecan is a pericellular HS proteoglycan encoded by the gene HSPG2 [43]. It has a molecular weight of 470 kDa and consists of a core protein with five domains and three HS chains attached to domain I and V [44,45]. Although downregulated in human atherosclerosis [46], it is the most prominent HS proteoglycan in the sub-endothelial matrix and an inhibitor of early intimal hyperplasia [47,48,49], suggesting an important role for this proteoglycan in the development and progression of atherosclerosis. Perlecan is located in the basement membrane [44] and can inhibit smooth muscle cell proliferation [44,47,50]. It does, however, have a pro-angiogenic effect on endothelial cells by enhancing cellular interactions with growth factors such as fibroblast growth factor (FGF) and vascular endothelial growth factor (VEGF) [51,52].

Perlecan has an inhibitory effect on platelet activity [48,53]. HS chains negatively regulate platelet function by binding to immunoreceptor tyrosine-based inhibition motif (ITIM)-containing receptor G6b-B on platelets, inhibiting platelet function and adhesion to the proteoglycan protein core via activation of inhibitory tyrosine phosphatases [52]. The degradation of perlecan observed in atherosclerosis may, therefore, propagate thrombus formation through loss of this inhibitory pathway.

### 3.2. Biglycan

Biglycan is a Class I small leucine-rich repeat proteoglycan (SLRP) located in the vascular wall, comprising a protein core and two GAG chains of CS or DS [31]. It is synthesized by a variety of cells, including endothelial cells, fibroblasts, and smooth muscle cells [39]. Biglycan can promote apposing protective and disease-promoting actions in the arterial wall—it associates with elastin and plays a role in elastin fiber assembly, increasing the structural integrity of the artery wall [31]. Additionally, biglycan localizes to areas of lipid deposition and may play a role in lipoprotein retention [54], promoting LDL trapping and subsequent oxidation and promotion of plaque development.

Biglycan has a pro-angiogenic effect on endothelial cells [55] through its ability to bind and concentrate VEGF-A, increasing growth factor concentrations in the sub-endothelial matrix [56]. It also supports smooth muscle cell migration by reducing focal adhesion kinase (FAK) levels and vinculin gene expression in cells [38], negatively regulating the formation of focal adhesions while suppressing proliferation by counteracting the effects of PDGF [37]. Biglycan exhibits pro-inflammatory effects on macrophages through TLR-2, TLR-4, and co-receptor CD14 activation [42,57]. Biglycan may therefore promote plaque formation and promote rupture through the potentiation of lipoprotein trapping, increase in inflammation, and inhibition of SMC proliferation, all features of rupture-prone plaques [9,10].

The role of biglycan in thrombosis has not been investigated in detail, however, several studies indicate a potential regulatory role, with a reduction in biglycan associated with increased thrombotic risk in fetal growth restriction [58] and increased platelet activation in mice [39]. Plasma levels of biglycan are elevated both in patients with acute coronary syndrome and apoE-/- mice [39]. Evidence from apo-/-/biglycan double deficient mice indicates that this is a protective mechanism and that biglycan negatively regulates thrombin generation by binding to and activating thrombin inhibitor heparin co-factor II (HCII) [59,60]. Collectively, these findings suggest an antithrombotic role for biglycan, indirectly suppressing platelet activity and the propagation of atherothrombosis. However, platelets only weakly adhere to biglycan [61], and the direct effect of biglycan on platelet function is yet to be determined.

### 3.3. Decorin

Decorin is a SLRP, structurally similar to biglycan, possessing a protein core but only one GAG chain of CS or DS [62]. Under physiological conditions, it is confined to the adventitia, but accumulates in atherosclerosis [35], predominantly in rupture prone plaques and in association with collagen I and biglycan [9,10,31]. Decorin binds to collagen I via the protein’s C-terminal [31,63,64], either directly [65] or indirectly through its GAG chain [66], and has a role in collagen crosslinking [43,62]. Decorin has a diverse interactome [67] and can modulate cellular behavior through direct activation of surface receptors, sequestration of growth factors, and by acting as a pan-receptor tyrosine kinase inhibitor. The sequatration of growth factors, including transforming growth factor (TGF)-β [35,68] and PDGF [69,70], has been shown to negatively regulate SMC growth and proliferation, a mechanism shown to prevent intimal hyperplasia after arterial injury [71]. Stimulation of the epidermal growth factor receptor (EGFR) pathway, resulting in phosphorylation of MAPK and activation of p21 [72], a suppressor of cell growth, has also been suggested; however, other studies suggest that decorin stimulation of EGFR can inhibit and down-regulate EGFR. Decorin promotes a migratory phenotype in smooth muscle cells [38] and has also been shown to promote SMC calcification [40]. In endothelial cells, however, decorin can increase proliferation and angiogenesis [64,73,74] as well as autophagy via VEGFR2 [75].

As a Class I SLRP, it is unsurprising that the effects of this proteoglycan on platelets and platelet-derived mediators share similarities with that of biglycan. Resting platelets bind decorin weakly [61], with adhesion facilitated by DS binding receptors on platelets, specifically integrin α2β1 [76]. Platelets can also interact with the protein core of decorin in a Mg^2+^-dependent mechanism [36]. Interaction of α2β1 with decorin results in tyrosine phosphorylation and activation of platelet Syk and phospholipase C (PLC) gamma, mediating shear-dependent platelet activation and aggregation under flow [77]. However, evidence suggests that this weak interaction protects platelets from a more thrombogenic high-affinity interaction with collagen, achieved by decorin masking [64], collagen binding sites, and preventing platelet adhesion and activation. Similar to biglycan, decorin can also indirectly reduce platelet reactivity by binding HCII and reducing thrombin generation [73] in blood. Findings from this study suggest that decorin may indirectly limit platelet activation via the inhibition of thrombin. Collectively, these findings indicate a role for decorin in platelet adhesion but not activation in atherothrombosis.

### 3.4. Versican

Versican is a large extracellular hyalectan encoded by the gene CSPG2 [43,47,51], and comprises an N-terminal G1 domain, which binds hyaluronan [47] and thrombospondin-1 [78], a GAG attachment region and an adhesive lectin-binding G3 domain [47,79,80,81]. At least five isoforms of versican exist due to alternate exon splicing [31]. Alternative-RNA splicing of exons 7 and 8, which encode versican GAG attachment sites α and β, respectively, regulate versican structure [47,82,83]. Isoform V0 contains both Exon 7 and 8, V1 contains Exon 8, V2 contains Exon 7, and V3 has no GAG attachment sites [79]. V4, the most recently discovered isoform, has been found in human breast cancer and contains a portion of the GAG β chain. Versican is degraded by the A disintegrin and metalloprotease with thrombospondin motifs (ADAMTS) family, MMPs, and serine proteases, such as plasmin [84,85,86]. ADAMTS-1 cleaves versican V0 and V1 isoforms [87] close to the N-terminal of the β domain, producing bioactive versikine [86], a pro-inflammatory and pro-apoptotic N-terminal fragment [88,89]. Versikine can induce the expression of inflammatory cytokines interleukin (IL)-1β and IL-6, counteracting the effects of the tolerogenic V1 isoform observed prior to proteolysis [90]. The expression of each versican isoform is linked to the cell phenotype [91,92], and each isoform holds a different function, which may be dependent on its location in the body [80,93]. V2, for example, is an isoform found predominantly in the human brain [94] where it has a role in matrix assembly and inhibition of axonal growth [95]. It is also found, however, in the vascular wall [61], where it is highly expressed during intimal thickening and endothelial to mesenchymal transition (EMT) [96], and has a role in endothelial angiogenesis. Versican expression is upregulated under pathological conditions, V1 and V3 being the most common isoforms found in smooth muscle cells [43] with opposing roles in the vasculature. V1 promotes cell proliferation and migration, whereas V3 suppresses cell activity and has an overall anti-inflammatory effect [97]. V3 expression reduces inflammation and monocyte adhesion and accumulation by NFκB signaling [98] in murine atherosclerosis [99] while increasing ECM elastic fiber assembly [31,100] via the regulation of TGF-β signaling [98]. V3 also promotes a differentiated, anti-inflammatory phenotype in smooth muscle cells [101], further establishing it as an endogenous atheroprotective agent in the vasculature.

Versican is a prominent proteoglycan found at the plaque–thrombus interface in plaque erosion, yet its role in thrombosis has not yet been fully established. V1 and V2 isoforms have been shown to promote platelet adhesion but not aggregation [9,61], potentially due to the ability of the G1 domain of versican to bind to thrombospondin-1 [78] and regulate thrombin-mediated platelet activation. Interestingly, the interaction between platelets and versican does not involve fibrinogen, vWF, or any of their known platelet receptors. It is charge-dependent and specific [61], suggesting a role for this proteoglycan in the initial stages of thrombosis and/or a platelet-dependent regulatory mechanism involving versican synthesis and degradation within atherosclerotic plaques. Indeed, PRP and PDGF have both been shown to stimulate versican synthesis and GAG elongation [102]. PDGF-mediated activation of endothelial cell ERK signaling pathways [103] and phosphorylation of Akt are associated with cell survival and proteoglycan core protein synthesis [104]. Conversely, platelets may also play a role in versican degradation. Thrombin generation mediated by platelet PS exposure has been suggested to regulate proteolytic degradation of versican [105], and may either contribute to atherothrombosis by producing smaller, potentially pro-inflammatory fragments of versican similar to versikine, or inhibit thrombosis as an additional mechanism of thrombin mediated self-regulation. Furthermore, the G3 domain of versican can enhance coagulation by binding to the tissue factor pathway inhibitor-1 (TFPI-1) [81], but it also binds L- and P-Selectins [106] including P-Selectin glycoprotein ligand-1 (PSGL-1). Platelets expressing PSGL-1 [107] may, therefore, competitively bind to the G3 domain to promote versican degradation and inhibit versican-induced coagulation. Thrombus growth is promoted only in the presence of collagen [61], indicating that the role of versican and its individual isoforms in thrombosis is complex and future studies investigating thrombosis should take into consideration the contribution of other proteoglycans to the thrombotic process.

### 3.5. Hyaluronan

Hyaluronan is a large, negatively charged, non-sulphated GAG with unique hydrodynamic properties [108,109,110] and is abundant in all tissues, including the vascular ECM. It is synthesized by hyaluronan synthases (HAS)-1, -2, and -3 on the inner surface of the cellular membrane [111] before being extruded into the extracellular space through a pore [93,108,110,112]. Hyaluronan binds to the receptor for hyaluronan-mediated motility (RHAMM) and CD44, which regulate cell migration and endothelial cell adhesion and proliferation, respectively [108,113,114]. Hyaluronan binding to CD44 evokes clustering and the activation p38, ERK-1 and -2, Akt, and focal adhesion kinase (FAK) [93], regulating cell proliferation and adhesion, proteoglycan synthesis and inflammation [104,115,116,117]. Hyaluronan exists in fragments of differing molecular weight. High molecular weight (HMW) hyaluronan is found in healthy tissue and has a molecular weight of over 100 kDa [110]. However, upon tissue injury, hyaluronan is fragmented into different sizes, which demonstrate differential biological activity based on molecular weight [92,118]. Low molecular weight (LMW) hyaluronan (<20 kDa) binds TLRs and promotes signaling pathways involving interleukin receptor-associated kinases (IRAKs) and tumor necrosis factor receptor-associated factors (TRAFs), resulting in the activation of NFκB and the propagation of inflammation [11,93,114,119].

HMW and LMW hyaluronan have differential effects on cells. HMW hyaluronan inhibits angiogenesis and cell proliferation, coating cells in a glycocalyx with antioxidant and anti-inflammatory properties [93,110,111,116,120] and maintaining a healthy endothelium. Together with fibrin, it crosslinks and forms a network in the ECM, aiding fibroblast and smooth muscle cell migration [10,93,121,122]. The excessive production of HMW hyaluronan, however, stimulates the release of hyaluronidase from fibroblasts [120], which degrades hyaluronan into angiogenic, inflammatory LMW fragments [110,111]. These LMW fragments induce cellular expression of pro-inflammatory molecules IL-6 and TNF-α and upregulation of TLR-4 expression [114]. Hyaluronan and versican together form key components of the pericellular matrix [112]. Upon tissue injury, growth factors such as PDGF stimulate smooth muscle cells to preferentially synthesize hyaluronan and versican, forming cable-like structures promoting cell migration, cell proliferation, and leukocyte adhesion [93,97,112,123]. These structures, however, are localized separately to focal adhesions in the vascular wall [112], and as hyaluronan regulates cell phenotype [91,92], together with versican, this GAG may regulate cell responses under both physiological and pathological conditions. The interactions of hyaluronan with various cell types have been well documented, and as a key component of the ECM in plaque erosion [9,10,31], its contribution to arterial thrombosis must also be considered. HMW hyaluronan binds to platelets via the CD44 receptor [124,125], and has been shown to inhibit platelet activation, aggregation, and adhesion [61,120,126,127]. Although LMW hyaluronan is pro-inflammatory [114], research suggests that it has a role in preventing atherogenic cardiovascular events. Modification of HMW hyaluronan through heavy chain (HC) transfer by tumor necrosis factor-stimulated gene-6 (TSG-6) is pathological and leukocyte adhesive [41,128]. Research conducted by Petrey et al. [129] revealed that HYAL-2 contained in platelet granules degrades HMW HC hyaluronan into LMW hyaluronan, significantly reducing leukocyte migration into the sub-endothelial layer. Additionally, HYAL-2 and CD44 interact to catalyze the cleavage of HMW hyaluronan to LMW hyaluronan [111] and are elevated in patients with plaque erosion [5]. In the absence of HYAL-2, hyaluronan accumulates and remodels the ECM [130], a pathological event central to the development of atherosclerosis. Collectively, these findings indicate an important role for platelets in plaque erosion, where hyaluronan is present at the plaque–thrombus interface and few leukocytes infiltrate the vascular wall [9,10,31]. Recent evidence has also demonstrated that increased HYAL-2 in thrombi found on eroded lesions occurs concomitantly with increased local levels of TLR2 ligand hyaluronic acid [131]. Neutrophil expression of TLR2 from patients with eroded plaques was also shown to be elevated compared to patients with ruptured plaques, with enhanced MMP-9 release from neutrophils and endothelial cytotoxicity [131]. Data from this study provides the first in-human evidence that hyalaronic acid may promote endothelial dissociation in plaque erosion through the TLR2 neutrophil axis.

**Table 1 ijms-24-11854-t001:** A summary of the cellular effects of vascular proteoglycans.

Proteoglycan	Platelet	Endothelial Cell	Smooth Muscle
Perlecan	↓ activation via G6b-B [52]	↑ proliferation and angiogenesis [51]	↓ proliferation [44,47]
Biglycan	↓ activation (indirectly by reducing thrombin levels) [39,59,60]	↑ proliferation and angiogenesis [55,56]	↓ proliferation ↑ migration [38] ↑ autophagy [42]
Decorin	↑ adhesion via platelet α2β1 [76]	↑ proliferation and angiogenesis [55]	↓ proliferation [37]
	↑ activation via PLCγ [36,77]	↑ autophagy [75]	↑ migration [38]
	↓ platelet adhesion to collagen by masking collagen binding sites [64]		↑ calcification [132]
Versican	↑ adhesion (V1 and V2 isoforms) [9,61]	↑ angiogenesis [96]	↑ proliferation and migration (V1) [97] ↑ Differentiation and anti-inflammatory (V3) [101]
Hyaluronan	↓ platelet activation via CD44 (HMW) [61,126]	↑ adhesion, proliferation, and migration [108,113] ↓ proliferation and angiogenesis [93,110,116] ↓ inflammation (HMW) ↑ inflammation (LMW) [93,119]	↑ migration (HMW) [93,121,122]

## 4. Future Directions

There is increasing evidence that a more personalized approach to antiplatelet therapy could provide increased protection against ACS while reducing the risk of bleeding. Clear distinctions between plaque rupture and erosion are evident in terms of patient demographics, plaque ECM composition, and the structure and composition of the subsequent thrombi, indicating discrete pathological mechanisms. Proteoglycans, in particular, vary significantly in their expression depending on plaque phenotype and exert multicellular effects that influence the development and progression of atherosclerosis, as well as the subsequent adhesion and activation of platelets following plaque disruption. The advances in vascular imaging through OCT now make it possible to assess plaque phenotype and stratify patients. However, until there are established stratified approaches in treatment, this offers little value. Promise has been demonstrated by the EROSION study, which used OCT to identify patients with eroded plaques and assessed clinical outcomes at 12 months following management on antithrombotics alone, without stenting [133]. The majority (92.5%) of patients remained free of cardiovascular events [133], however, with safer, more efficacious antithrombotic therapy, there is potential to improve this further. Advances in GPVI targeting as a novel antithrombotic approach offers the potential of plaque-directed antithrombotic therapy, without the risk of bleeding [29]. Clinical trials are ongoing, however, the cost and route of administration of these agents may limit their clinical utility. Further research to understand how ECM components contribute to atherothrombosis, particularly those which are differentially expressed in “rupture-” verses “erosion”-prone plaques, is necessary to inform the development of novel personalized antithrombotics.

## Figures and Tables

**Figure 1 ijms-24-11854-f001:**
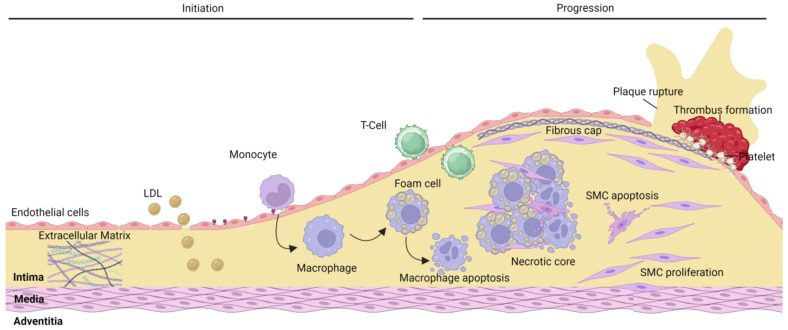
Atherosclerosis initiation and progression. The early development of atherosclerosis involves the uptake of LDL cholesterol into the intima and the activation of endothelial cells (ECs), leading to an upregulation in adhesion molecules and increased vascular permeability. Monocytes adhere to endothelial adhesion receptors and migrate into the intima, where they differentiate into macrophages. Scavenger receptors present on the surface of macrophages facilitate the uptake of LDL particles resulting in the transition to foam cells. T lymphocytes also migrate into the intima, where they regulate ECs, smooth muscle cells (SMCs), and innate immune cell function. As the plaque progresses, the fibrous cap can weaken and thin through matrix metalloprotease degradation and reduced capacity of SMCs to regenerate the ECM. Apoptosis of SMCs and macrophages results in a necrotic core comprised of cell debris, rich in lipids. Created using BioRender.

**Figure 2 ijms-24-11854-f002:**
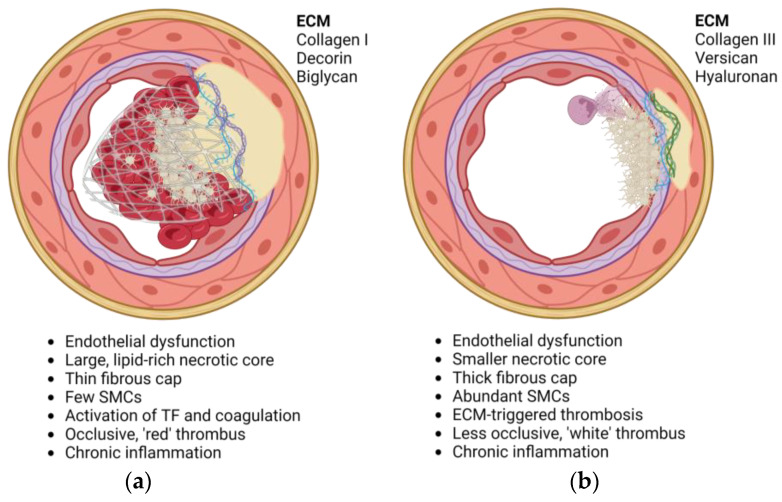
Schematic diagrams demonstrating the structure, composition, and risk factors associated with plaques phenotypic of (**a**) plaque rupture and (**b**) plaque erosion. Created using BioRender.

**Figure 3 ijms-24-11854-f003:**
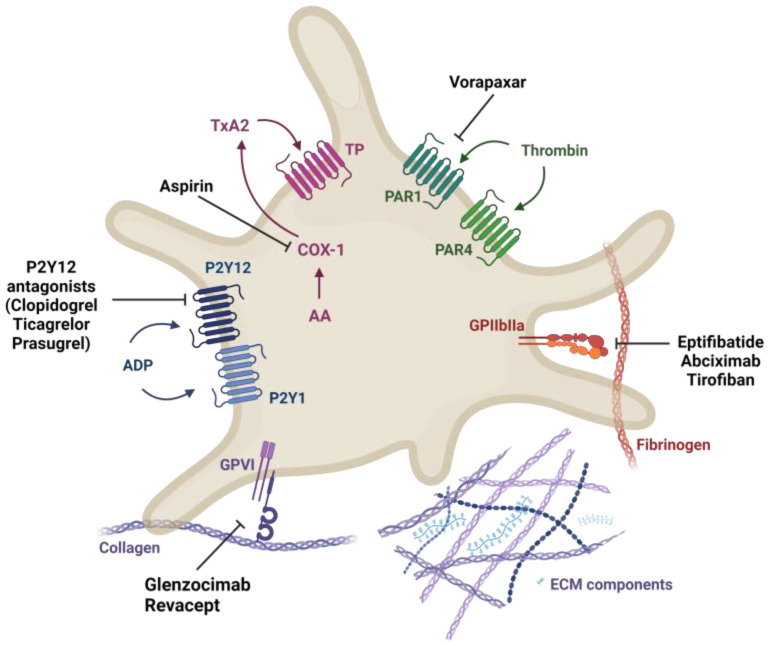
Antiplatelet drugs. Aspirin inhibits cyclo-oxygenase 1 (COX-1), a key enzyme involved in the conversion of arachidonic acid (AA) to thromboxane A2 (TXA2). Clopidogrel, prasugrel, and ticagrelor target the ADP receptor P2Y12 on the platelet surface. Vorapaxar inhibits thrombin protease-activated receptor 1 (PAR1). Abciximab, eptifibatide, and tirofiban inhibit platelet aggregation by blocking GPIIb/IIIa, preventing fibrinogen and von Willebrand factor binding. Glenzocimab and revacept are novel drugs currently in clinical trial, which interfere with GPVI-collagen binding. Created using BioRender.

## Data Availability

No new data were created or analyzed in this study. Data sharing is not applicable to this article.

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
