# Peer review of "The Contribution of Vascular Proteoglycans to Atherothrombosis: Clinical Implications"

_ijms, 2023, doi:10.3390/ijms241411854_

Round 1
Reviewer 1 Report
The authors perform an incredibly comprehensive evaluation of proteoglycans on ACS
The authors are to be praised for the novelty of the manuscript because few articles focus on proteoglycans. The authors should also be praised for the incredibly comprehensive and in-depth review performed. Furthermore, the manuscript is short and reads well.
I only have minor comments about this manuscript:
- The authors should mention the recent article Meteva D, Eur Heart J 2023 Jun 29;ehad379, provides first in-human evidence for the role of hialuran on distinct TLR2-mediated neutrophil activation in ACS due to plaque erosion, presumably triggered by elevated soluble hyaluronic acid. Toll-like receptor 2 (TLR2) expression was higher on neutrophils from plaque erosion than plaque rupture. TLR2 stimulation increased the release of active MMP9 from local plaque erosion-derived neutrophils, which also aggravated endothelial cell death independently of TLR2. Thrombi of plaque erosion patients exhibited more hyaluronidase 2 with concomitant increase in local plasma levels of the TLR2 ligand: hyaluronic acid
- The authors should also briefly mention the role of proteoglycans on other diseases outside ACS, such as heart failure (please quote J Am Coll Cardiol. 2015 Feb 3;65(4):378-388). The classic notion of extracellular sodium involves two compartments (intravascular and interstitial distribution); if there is excessive sodium intake, sodium will accumulate in the interstitium, water will follow to the interstitium and peripheral edema is created. Recent articles have demonstrated that there is a third compartment: proteoglycans in the interstitial tissue are heavily negatively charged with sulphate/phosphate groups, so they can bind sodium in a non-osmotic way. This means that proteoglycans can act as a buffer for sodium overload, that they can accept sodium without water following, that they can store the excess of sodium without inducing congestion. Only if proteoglycans are dysfunctional, then congestion will occur. Interestingly, sodium overload changes the negatively charged sulfate residues in the proteoglycans, which results in proteoglycan dysfunction while spironolactone prevented these harmful effects of Na+ (Pflugers Arch, 462 (2011), pp. 519-528). This concept of proteoglycan dysfuncion originally shown in heart failure can probably also be applied to plaque erosion.
- Line 126: Before describing the effect of P2Y12 inhibitors, please provide one sentence summarizing the effect of aspirin, and quote the relevant article Am J Cardiol. 2021 Apr 1;144 Suppl 1:S2-S9
Author Response
We would like to thank the reviewers for taking the time to review our manuscript and providing such positive comments. We have now made the appropriate changes following the valuable suggestions from the reviewer.
- Reference has now been made to the important recent study by Meteva et al., 2023 (L359-365)
- We have briefly mentioned the role of proteoglycans in other diseases eg heart failure and included the suggested reference (L158-162)
- We have included a brief summary of how aspirin works and included the suggested reference.
Reviewer 2 Report
“The Contribution of Vascular Proteoglycans to Atherothrombosis: Clinical Implications”
In this review, Drysdale A. et al. describe the effects of proteoglycans on cellular behavior and platelet response, highlighting their role in atherothrombosis and how this knowledge may suggest a targeted approach to antithrombotic treatments improving clinical outcomes in cardiovascular diseases.
This manuscript is very well written, the writing is well organized with very clear figures and up to date references.
I have only minor points.
Line 17 delete point before moreover.
Line 43 delete from.
There is an extra bulleted list in the part B of Figure 2
Line 248 Add a point before Findings.
Line 343 Add a space after considered.
Author Response
We thank the reviewer for taking the time to read our manuscript. All of the minor changes have been made.